# Discordance of HER2-Low between Primary Tumors and Matched Distant Metastases in Breast Cancer

**DOI:** 10.3390/cancers15051413

**Published:** 2023-02-23

**Authors:** Katrin Almstedt, Lisa Krauthauser, Franziska Kappenberg, Daniel-Christoph Wagner, Anne-Sophie Heimes, Marco J. Battista, Katharina Anic, Slavomir Krajnak, Antje Lebrecht, Roxana Schwab, Walburgis Brenner, Wolfgang Weikel, Jörg Rahnenführer, Jan G. Hengstler, Wilfried Roth, Annette Hasenburg, Kathrin Stewen, Marcus Schmidt

**Affiliations:** 1Department of Obstetrics and Gynecology, University Medical Center Mainz, 55131 Mainz, Germany; 2Department of Statistics, TU Dortmund University, 44227 Dortmund, Germany; 3Institute of Pathology, University Medical Center Mainz, 55131 Mainz, Germany; 4Leibniz-Research Centre for Working Environment and Human Factors at the TU Dortmund (IfADo), 44139 Dortmund, Germany

**Keywords:** HER2-low, HER2-zero, HER2 overexpression, HER2 dynamics, de-novo metastasis, antibody-drug conjugate

## Abstract

**Simple Summary:**

Novel antibody-drug conjugates (ADCs) show efficacy in advanced breast cancer with low HER2 levels. Little is known about the discordance of low HER2 levels between the primary tumor and distant metastases. The clinical relevance of discordance between the primary tumor and metastases prompted us to investigate the differences in HER2 expression between primary tumors and distant metastases, particularly in the HER2-negative (HER2-low and HER2-zero) primary breast cancer cohort. Our results show a relevant discordance of HER2-low status between primary tumors and their corresponding distant metastases.

**Abstract:**

We examined differences in HER2 expression between primary tumors and distant metastases, particularly within the HER2-negative primary breast cancer cohort (HER2-low and HER2-zero). The retrospective study included 191 consecutive paired samples of primary breast cancer and distant metastases diagnosed between 1995 and 2019. HER2-negative samples were divided into HER2-zero (immunohistochemistry [IHC] score 0) and HER2-low (IHC score 1+ or 2+/in situ hybridization [ISH]-negative). The main objective was to analyze the discordance rate between matched primary and metastatic samples, focusing on the site of distant metastasis, molecular subtype, and de novo metastatic breast cancer. The relationship was determined by cross-tabulation and calculation of Cohen′s Kappa coefficient. The final study cohort included 148 paired samples. The largest proportion in the HER2-negative cohort was HER2-low [primary tumor 61.4% (*n* = 78), metastatic samples 73.5% (*n* = 86)]. The discordance rate between the HER2 status of primary tumors and corresponding distant metastases was 49.6% (*n* = 63) (Kappa −0.003, 95%CI −0.15–0.15). Development of a HER2-low phenotype occurred most frequently (*n* = 52, 40.9%), mostly with a switch from HER2-zero to HER2-low (*n* = 34, 26.8%). Relevant HER2 discordance rates were observed between different metastatic sites and molecular subtypes. Primary metastatic breast cancer had a significantly lower HER2 discordance rate than secondary metastatic breast cancer [30.2% (Kappa 0.48, 95%CI 0.27–0.69) versus 50.5% (Kappa 0.14, 95% CI −0.03–0.32)]. This highlights the importance of evaluating potentially therapy-relevant discordance rates between a primary tumor and corresponding distant metastases.

## 1. Introduction

Overexpression or amplification of human epidermal growth factor receptor 2 (HER2) characterizes a molecular subtype of breast cancer that progresses rapidly and has a poor prognosis [1,2]. However, with the advent of targeted therapies against HER2 such as the monoclonal antibody trastuzumab, the original prognostic disadvantage of HER2 positivity has been transformed into a clinically relevant predictive advantage [3]. In advanced HER2-positive breast carcinoma, survival was further prolonged by pertuzumab [4]. In the event of progression, tyrosine kinase inhibitors such as lapatinib or tucatinib showed efficacy in HER2-positive breast carcinoma [5,6]. In addition, antibody-drug conjugates (ADC) provided a further improvement not only in progression-free survival (PFS) but also in overall survival (OS) in advanced HER2-positive breast carcinoma [7,8].

Therefore, it was obvious to use anti-HER2 therapies also in early breast carcinoma. Indeed, the use of trastuzumab resulted in a statistically significant prolongation of overall survival [9]. Surprisingly, some of the patients who participated in the original trastuzumab trials and were ultimately HER2-negative at central HER2 reassessment benefited from trastuzumab [10,11]. Based on these findings, a large phase III trial was conducted in 3270 women, but it clearly showed that trastuzumab was not beneficial for patients without IHC 3+ or ISH-enhanced breast cancer [12]. In HER2-positive early breast cancer, the addition of pertuzumab led to a relevant increase in pathologic complete response rates (pCR) and an improvement in disease-free survival (DFS) [13,14]. The prolongation of DFS was further increased using the tyrosine kinase inhibitor neratinib after the completion of trastuzumab-based therapy in HER2-positive patients [15]. Furthermore, the ADC T-DM1 improved DFS in early HER2-positive breast cancer with residual disease after neoadjuvant trastuzumab-based treatment [16]. Overall, these HER2-targeted therapies represent tremendous progress for the 15–20% HER2-positive patients. Meanwhile, several retrospective studies have taken a closer look at the large group of HER2-negative breast cancer. Breast cancer showing HER2 protein expression without HER2 gene amplification could be divided into two separate groups (HER2-low [IHC 1+ or 2+ and ISH-negative] and HER2-zero [IHC 0]) with different prognosis or pCR after neoadjuvant chemotherapy [17,18,19,20,21,22]. However, these results could not be confirmed in several other studies [23,24,25,26,27,28,29,30,31,32,33,34,35,36]. Nevertheless, interest in HER2-low tumors has increased greatly due to the results of the DESTINY-Breast04 trial, which demonstrated the superiority of trastuzumab-deruxtecan (T-DXd) over physician′s choice chemotherapy in patients with advanced HER2-low breast cancer [37]. The prolongation of PFS (10.1 months vs. 5.4 months; hazard ratio [HR] 0.51; *p* < 0.001) and OS (23.9 months vs. 17.5 months; HR 0.64; *p* = 0.003) was both statistically significant and clinically relevant. These compelling results led to a rapid update of the American Society of Clinical Oncology (ASCO) guideline and a positive opinion of the Committee for Medicinal Products for Human Use (CHMP) of the European Medicines Agency (EMA) recommending the use of T-DXd in patients with HER2-low metastatic BC [38,39].

Until now, the only question regarding HER2 status was whether the tumor was HER2-positive or HER2-negative. However, the impressive data from T-DXd in HER2-low breast cancer highlights the importance of dividing the large group of HER2-negative patients. In principle, the well-known ASCO/College of American Pathologists (CAP) clinical practice guideline allows such a distinction [40]. However, potential difficulties such as tumor heterogeneity (clustered or mosaic type) or unusual staining patterns (moderate to intense but incomplete staining or carcinomas with limited strong HER2 overexpression) must be considered [41]. To address difficulties in distinguishing between HER2-low and HER2-zero, pathologists have already pointed out possible solutions for the assessment of immunohistochemical staining such as (1) application of the “magnification rule”, (2) staining pattern-circularity of membrane staining, and (3) percentage of tumor cells with HER2 expression [42].

In addition to these briefly outlined challenges for pathologists in distinguishing between HER2-low and HER2-zero, another fundamental problem is discordance between the primary tumor and corresponding metastases, since whenever possible, a recent metastatic biopsy is encouraged to guide therapy in advanced breast cancer. The problem of discordance between primary tumors and distant metastases arises when treating patients with metastatic disease with targeted therapies. The discordance of traditional HER2 dichotomy (positive or negative) between primary breast cancer and distant metastases is well established. Among others, Grassini et al. reviewed the phenomenon of HER2 conversion between primary breast tumors and relapsed/distant metastatic [41]. While early studies described a wide variability in HER2 discordance rates (0–44%), several meta-analyses showed discordance rates ranging from 7.8% to 13.7% [43,44,45]. Most commonly, conversion from HER2-positive to HER2-negative was observed, which is clinically important in both advanced and early breast cancer. In neoadjuvant studies, a loss of HER2 expression from a therapy-naïve primary tumor and the post-neoadjuvant residual tumor was described with a prognostic disadvantage [46,47,48,49,50,51].

However, less is known about the discordance of HER2-low between primary tumors and distant metastases. Thus, Tarantino and coworkers demonstrated a relevant discordance in HER2 expression between PTs and their associated metastases: 44% of HER2-zero PTs had an elevated HER2 score on biopsy, and 22% of HER2-low PTs became HER2-zero tumors [32]. Miglietta et al. reported an overall rate of HER2 discordance of 38.0%, with most transitioning from HER2-zero to HER2-low (15%) and from HER2-low to HER2-zero (14%) [52]. This discordance rate is clinically relevant to the use of ADCs and prompted us to investigate the discordance rate in 148 paired samples (primary breast tumor and distant metastasis), focusing on (i) molecular subtype, (ii) distant metastasis site, and (iii) differences between primary metastatic breast cancer (PMBC) and secondary metastatic breast cancer (SMBC).

## 2. Materials and Methods

### 2.1. Study Cohort

The certified breast cancer center of the University Medical Center Mainz has been prospectively documenting clinic-pathological characteristics as well as therapies of all treated breast cancer patients. This database was searched for patients with metastatic breast cancer between 15.06.1995 and 10.10.2019. We studied 191 consecutive paired samples of primary breast tumor and distant metastasis. Only solid distant metastases were considered. Paired samples without complete HER2 status (*n* = 31), with HER2 status equivocal (*n* = 4), or bilateral BC or secondary malignancy with different HER2 status (*n* = 8) were not eligible for this study (Figure 1).

PMBC was defined as the presence of metastasis at the time of diagnosis of the PT [53,54]. The median age at the time of initial breast cancer diagnosis was 53 years (range, 31–86 years). Table 1 provides an overview of the established clinico-pathologic prognostic factors in the final study cohort (*n* = 148).

### 2.2. Immunohistochemistry (IHC) and In Situ Hybridization (ISH)

Immunohistochemical analyses and in situ hybridization were performed on 3 µm thick sections of paraffin-embedded formalin-fixed tissues according to standard procedures. HER2 was scored from 0 to 3+ [40]. HER2 2+ cases (*n* = 22) were further classified as amplified or non-amplified by either fluorescence in situ hybridization (FISH) (Her2 FISH pharmDX kit, Dako) or chromogenic in situ hybridization (CISH) (Ventana Her2 dual ISH assay, Roche). HER2 2+ tumors with amplification of HER2 and 3+ tumors were classified as HER2-positive. The HER2-negative cohort was defined as 0, 1+, and 2+ without amplification of HER2. HER2-low tumors included all 1+ and 2+ tumors without amplification of HER2. Tumors with a HER2 score 0 were classified as HER2-zero [41,42]. Hormone receptor status was positive if tumor cells showed nuclear expression of either the estrogen receptor (ER) and/or the progesterone receptor (PR), the cut-off being defined as 1% of tumor cells [55].

The study was approved by the Ethics Committee of the Rhineland-Palatinate Medical Association, Germany (2021-15657). Written informed consent was obtained from all patients, and all clinical investigations were performed according to ethical and legal standards and in compliance with the Declaration of Helsinki. This study follows the Strengthening the Reporting of Observational Studies in Epidemiology (STROBE) reporting guideline [56].

### 2.3. Statistical Analysis

The main objective was to evaluate HER2 expression differences between primary tumor and distant metastasis, particularly in the HER2-negative (HER2-low and HER2-zero) primary breast cancer collective. Secondary objectives of our analysis were (i) that discordance rates differ according to the molecular subtype of the primary tumor, (ii) that discordance rates differ according to the site of distant metastasis, and (iii) the discordance rate in PMBC is lower than in SMBC. The relationship between the different categorical variables was determined by cross-tabulation. Comparisons between different HER2 status (primary tumor and metastasis) were calculated by Cohen’s Kappa coefficient. Relationships between HER2 status for primary tumor and clinico-pathological parameters were assessed by cross-tabulation and using Pearson’s chi-squared test (χ^2^ test). Statistical analyses were performed using the SPSS statistical software program, version 27.0 (SPSS Inc., Chicago, IL, USA), and the statistical language R, version 4.1.2 [57]. Patients’ characteristics were given in absolute and relative numbers. All *p*-values are two-sided. Because no correction was made for multiple testing due to the exploratory nature of the study, these are descriptive measures that should be interpreted with caution.

## 3. Results

### 3.1. Patient Population

The final study cohort included 148 paired samples. Primary tumors were divided into 127 (85.8%) HER2-negative samples [49 HER2-zero (38.6%) and 78 HER2-low (61.4%)] and 21 (14.2%) HER2-positive samples (Figure 1). One-hundred and seven (72.3%) primary tumors showed a luminal-like phenotype, 21 (14.2%) were HER2-positive, and 20 (13.5%) had a triple-negative phenotype. PMBC occurred in 35.8% (*n* = 53), and more frequently in the HER2-low (58.5%, *n* = 31) than in the HER2-zero cohort (15.1%, *n* = 8). The median time to first metastasis was 25 months (range 0–150). The median time between diagnosis of metastatic disease and biopsy was one month (range 0–131). Metastases were located in the liver (*n* = 50, 33.8%), bone (*n* = 38, 25.7%), skin/soft tissue (*n* = 18, 12.2%), central nervous system (CNS) (*n* = 15, 10.1%), other sites (*n* = 13, 8.8%), lung/pleura (*n* = 9, 6.1%), and lymph nodes (*n* = 5, 3.4%). Seventy-nine (53.4%) patients received adjuvant endocrine therapy and 72 (48.6%) neo-/adjuvant chemotherapy. A small proportion of patients were treated with adjuvant anti-HER2 therapy (*n* = 9, 6.1%). At the time of metastatic biopsy, 35.1% (*n* = 52) of patients were receiving endocrine therapy, 30.4% (*n* = 45) chemotherapy, and/or 8.8% (*n* = 13) anti-HER2 therapy for metastatic disease. Compared with HER2-zero and HER2-positive phenotype, HER2-low was significantly less frequently diagnosed in larger tumors (>T2) (HER2-low 66.0% vs. HER2-zero 77.1% and HER2-positive 85.6%, *p* = 0.032) and poorly differentiated tumors (G3) (HER2-low 33.3% vs. HER2-zero 49.0% and HER2-positive 60.0%, *p* = 0.049). HER2-low status was more common in tumors with higher Ki-67 (>20%) compared with HER2-zero (82.7% vs. 64.3%). However, higher Ki-67 levels were most frequently found in the HER2-positive cohort (100%) (*p* = 0.022). Low HER2 was significantly more common in luminal-like tumors than in triple-negative tumors (88.5% vs. 11.5%), while conversely, HER2-zero was more common in triple-negative tumors (22.4% vs. 11.5%) (*p* < 0.001). Additional tumor and patient characteristics are listed in Table 1.

### 3.2. Change of HER2 Status between Primary Breast Cancer and Metastasis

In the HER2-negative cohort, the HER2-low phenotype represented the largest group [primary tumor 61.4% (*n* = 78), metastatic samples 73.5% (*n* = 86)]. Discordance in HER2 status between the primary tumor and the matched metastatic biopsy was 49.6% (*n* = 63) (Kappa −0.003, 95%CI −0.15–0.15). Development of the HER2-low phenotype (HER2-zero to HER2-low or HER2-low to HER2-zero) was most common (*n* = 52, 40.9%), especially with enrichment to HER2-low (*n* = 34, 26.8%) (Figure 2 and Appendix A).

In the entire cohort (*n* = 148), HER2 discordance was 43.2% (*n* = 64) (Kappa 0.270, 95%CI 0.14–0.41). Most frequently, an evolution from HER2-zero to HER2-low phenotype was observed (*n* = 34, 23.0%). Within the HER2-zero cohort, this represented a switch of 69.4% from HER2-zero to HER2-low. A change from HER2-low to HER2-zero occurred in 12.2% (*n* = 18). Considered for the HER2-low cohort alone, a switch from HER2-low to HER2-zero resulted in 23.1%. The HER2-positive cohort showed the greatest stability, with a discordance of 4.8% (*n* = 1 of 21) (Appendix A and Appendix A). When additional metastatic biopsies were performed, a discordance rate of 57.9% was observed compared with the previous biopsy (Table 1). Again, the most common finding was a change from HER2-zero to HER2-low (15.8%; within the HER2-zero cohort: 60.0%).

### 3.3. Change of HER2 Status in Different Metastatic Sites

The proportion of HER2-low did not differ significantly between the different metastatic sites (*p* = 0.349) (Table 1). In the HER2-negative population, a relevant HER2 discordance rate was observed between the different metastatic sites (bone: Kappa 0.022, -0.230–0.273; liver: Kappa 0.048, −0.195–0.291; skin/soft tissue: Kappa 0.082, −0.363–0.527, lymph node: Kappa 0.000; CNS: Kappa −0.250, −0.606–0.106; others: Kappa −0.467, −0.980–0.047). Only pulmonary/pleural metastases showed absolute concordance with the primary breast tumor (Kappa 1.0, 95%CI 1.0–1.0). An increase from HER2-zero to HER2-low at metastatic biopsy was most frequently detected, excluding bone metastases. A switch to HER2 positive has been observed especially in CNS metastases (*n* = 2, 20.0%) (Appendix A and Figure 3).

Similarly, there was a relevant change in HER2 expression in the entire cohort (Appendix A and Appendix A).

### 3.4. Change of HER2 Status in Different Molecular Subtypes

HER2 discordance was observed according to the molecular subtype, in the Luminal A/B cohort (Kappa −0.044, −0.202–0.114) and in triple-negative breast cancer (Kappa 0.107, −0.247–0.461). In both subcohorts, a switch from HER2-zero to HER2-low was most frequently detected (Luminal A/B *n* = 28, 26.2%; Triple-negative *n* = 6, 30.0%) (Appendix A and Appendix A).

### 3.5. Change of HER2 Status in Primary vs. Secondary Metastatic Breast Cancer

PMBC was evidenced in 35.8% (*n* = 53). HER2-low represented the largest proportion in both the primary tumor (58.5%, *n* = 31) and matched de-novo metastases (54.7%, *n* = 29). The de-novo cohort showed a higher prevalence of HER2-low expression in PT (58.5%, *n* = 31) than in the relapsed collective (49.5%, *n* = 47). The HER2-low phenotype was represented more frequently in secondary metastases than in de-novo metastases (60.0% vs. 54.7%) (Appendix A). The discordance rate was lower in the PMBC than in the SMBC cohort [30.2% (Kappa 0.48, 95%CI 0.27–0.69) vs. 50.5% (Kappa 0.14, 95% CI −0.03–0.32)]. In the de-novo cohort, the increase almost corresponded to the loss of HER2 expression (11.3% vs. 13.2%), whereas in the SMBC cohort, the change from HER2-zero to HER2-low clearly predominated (29.5% vs. 11.6%) (Figure 4, Appendix A).

## 4. Discussion

In our retrospective analysis, we showed a relevant discordance rate of HER2 status between primary breast cancer and distant metastases, with the conversion from HER2-zero to HER2-low observed most frequently. Recently published studies have addressed the heterogeneity of HER2-negative breast cancer, focusing on the HER2-negative cohort [17,19,20,28]. The need to relativize the traditional dichotomization between HER2-positive and HER2-negative appeared at the latest with the results of the DESTINY-Breast04 trial, which demonstrated the superiority of trastuzumab-deruxtecan (T-DXd) vs. chemotherapy of the physician’s choice in patients with advanced HER2-low breast cancer [37]. In addition, other HER2-targeting ADCs like trastuzumab duocarmazine showed promising activity in early studies [58]. In our previous study, we reported a rate of 48.3% HER2-low tumors, which was within the range of approximately half of HER2-negative breast cancer patients (31.0% to 60.6%) reported by Prat and coworkers in a recent review article [20,58]. Since 80–85% of all breast cancer tumors have a HER2-negative phenotype, which was 85.2% in our study, a better understanding of this cohort has potential therapeutic implications for the majority of breast cancer patients. In this context, the aspect of the evolution of HER2 expression from early to advanced breast cancer is important. In the present study, we demonstrated a discordance rate between HER2 status of primarytumors and associated distant metastases within the HER2-negative cohort of 49.6% (Kappa −0.003, 95%CI −0.15–0.15). The development of HER2-low occurred frequently (40.9%), particularly with a switch from HER2-zero to HER2-low (26.8%). Discordance rates from our study were slightly higher than in other studies (38.0% and 40.9%, respectively [52,59]. However, all studies showed an increase in HER2 expression from HER2-zero to HER2-low during disease progression. Our results of additional metastatic biopsies compared with initial metastatic biopsies point in the same direction, with a discordance rate of 57.9%. There are several hypotheses for low HER2 stability (e.g., genetic drift and clonal evolution during tumor progression, intratumoral heterogeneity, and the selective effect of administered therapies) leading to the enrichment of HER2 expression [41,60,61,62,63,64,65,66]. Our study cohort showed heterogeneity in terms of time to metastasis, with a significantly longer time in the HER2-zero cohort than in the HER2-low cohort [median 44 months (0–150) vs. 14 months (0–121)] and in terms of systemic treatments given. Neo-/adjuvant chemotherapy was significantly more common in the HER2-zero than in the HER2-low cohort (71.4% vs. 38.5%, *p* < 0.001). Both aspects may have an impact on the increase of HER2-low from a primary tumor to distant metastases. In our study, there were no significant differences in HER2-low expression depending on the metastatic site (*p* = 0.349). Comparable to our results, Tarantino and coworkers also found no significant difference in HER2-low expression at different metastatic sites (*p* = 0.88), even when they divided biopsy sites into visceral (liver, lung, and pleura) and nonvisceral (skin and soft tissues, lymph nodes, bone, other) (*p* = 0.56) [32]. Miglietta et al. examined locoregional recurrences in addition to HER2-low prevalence in various distant metastases, also with similar results for HER2-low [52]. However, in Miglietta′s cohort, a significant difference in discordance rates was observed between the different metastases (*p* = 0.001), with the greatest HER2 instability in liver and bone metastases and the greatest concordance in lung and CNS metastases. Lung/pleural metastases also had the strongest concordance in our study. In the HER2-negative cohort, discordance rates ranged from 40.9% (liver) to 80.0% (CNS). The HER2 score changed from HER2-zero to HER2-low most frequently at metastatic biopsy, except for bone metastases, which is in contrast to the results of Lin et al. [66]. Overall, the aspects of similar HER2 expression levels at different metastatic sites and the different discordance rates are of clinical importance when a metastatic site has to be selected for biopsy to decide on targeted therapies that are also effective in HER2-low tumors. Another objective of our study was to analyze whether discordance rates depend on the molecular subtype of the primary BC. Recently published studies that examined discordance rates as a function of the molecular subtype of the primary tumor showed that HER2-low was more common in HR-positive tumors than in triple-negative tumors [22,32,52,59,67,68]. Similarly, the HR-positive subtype was associated with a higher discordance rate, most frequently switching from HER2-zero to HER2-low [32,52]. In our study, HER2-low was also significantly more common in luminal-like tumors than in triple-negative tumors, although the difference was much greater than in the above studies. 

A particular aspect of our study was the evaluation of PMBC, which we assessed for HER2-low prevalence and discordance rates. Compared to SMBC, PMBC had a significantly lower discordance rate between the PT and matched metastases [30.2% (Kappa 0.48, 95%CI 0.27–0.69) versus 50.5% (Kappa 0.14, 95% CI −0.03–0.32)]. While an increase in HER2 expression between primary tumor and distant metastases was observed across the cohort in our study, this trend was not observed in PMBC, supporting the hypothesis of HER2 enrichment during tumor progression. Overall, not all studies clearly addressed PMBC [52], excluded de-novo tumors, or applied various de-novo definitions (e.g., <6 months) [32,59]. Therefore, a comparison between our results and those of other studies is difficult. Although the de novo cohort of our study was small (*n* = 53), the aspect of low HER2 instability should also receive attention in de novo metastatic BC. In this regard, further studies with larger study cohorts are needed.

Our study has several limitations, such as the retrospective and unicenter setting. Another limitation is the lack of a central pathology assessment given the low interobserver reproducibility, especially in HER2-low and HER2-zero [69]. In addition, we did not investigate a possible prognostic impact of HER2 discordance between the primary tumor and the corresponding distant metastases. However, the consecutive inclusion of matched pairs is a strength. In addition, we examined the impact of both the site of distant metastases and molecular subtypes and took a closer look at the PMBC group.

## 5. Conclusions

In summary, we have shown in our study that there is a significant discordance rate between HER2-negative primary breast cancer and the corresponding metastases, with HER2-zero tumors more likely to become HER2-low tumors at advanced stages. In contrast, there is less discordance in PMBC. This evolution of HER2 expression is a clinically relevant rationale for metastatic biopsy and offers patients the opportunity to receive an effective treatment such as trastuzumab-deruxtecan.

## Figures and Tables

**Figure 1 cancers-15-01413-f001:**
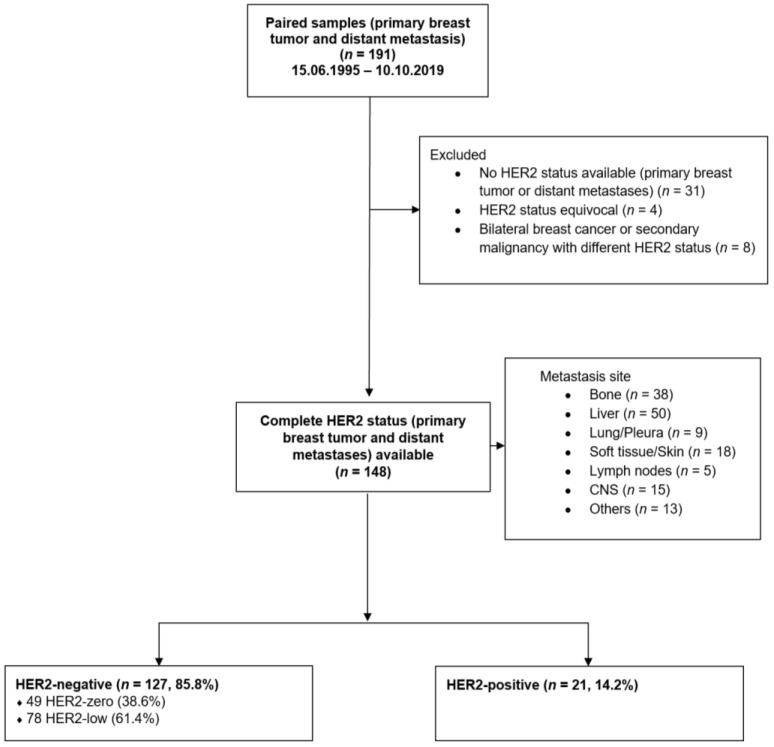
Patient enrolment.

**Figure 2 cancers-15-01413-f002:**
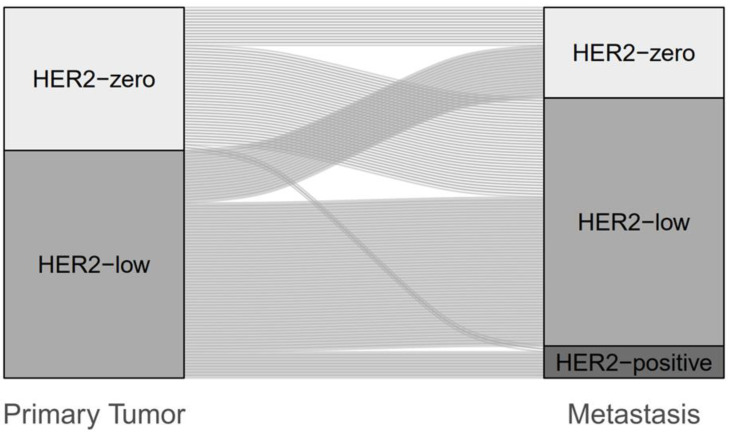
Change of HER2 status between primary tumor and metastasis in the HER2-negative cohort (*n* = 127).

**Figure 3 cancers-15-01413-f003:**
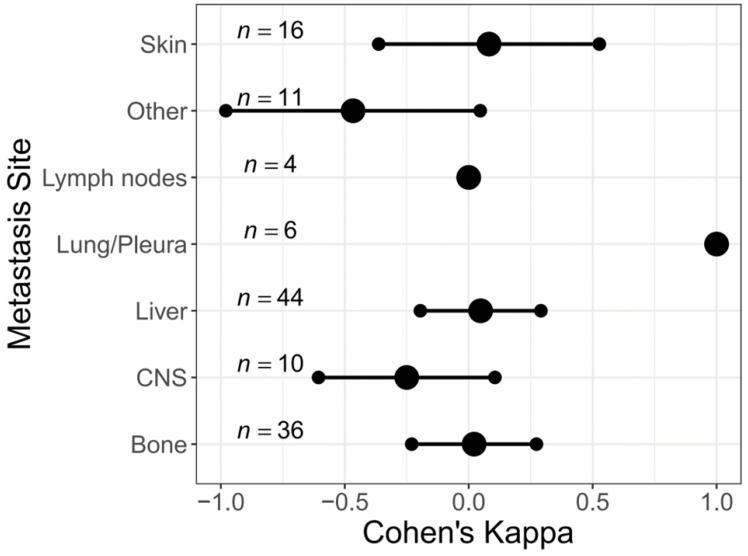
Change of HER2 status in different metastatic sites in the HER2-negative cohort (*n* = 127).

**Figure 4 cancers-15-01413-f004:**
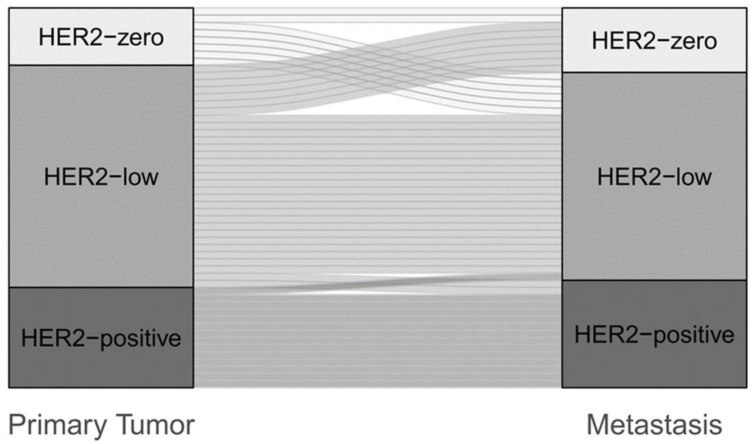
Change of HER2 status between primary tumor and metastasis in the de-novo cohort (*n* = 53).

**Table 1 cancers-15-01413-t001:** Clinico-pathological parameters at diagnosis (*n* = 148).

		Total Number of Patients (*n* = 148)	HER2-Negative (*n* = 127)		HER2-Positive (*n* = 21)	*p*-Value
			HER2-Zero (*n* = 49)	HER2-Low (*n* = 78)		
**Age at primary breast surgery**						0.087
	Median [years]	53	48	54	52	
	<50 years	66 (44.6%)	28 (57.1%)	29 (37.2%)	9 (42.9%)	
	>50 years	82 (55.4%)	21 (42.9%)	49 (62.8%)	12 (57.1%)	
**Histological subtype**						0.221
	Invasive carcinoma of no special type (NST)	115 (77.7%)	33 (67.3%)	63 (80.8%)	19 (90.5%)	
	Invasive lobular carcinoma	24 (16.2%)	12 (24.5%)	11 (14.1%)	1 (4.8%)	
	other	9 (6.1%)	4 (8.2%)	4 (5.1%)	1 (4.8%)	
**Tumor size**						0.032
	pT1	32 (22.2%)	11 (22.9%)	18 (24.0%)	3 (14.3%)	
	pT2	67 (46.5%)	30 (62.5%)	30 (40.0%)	7 (33.3%)	
	pT3/4	45 (31.3%)	7 (14.6%)	27 (36.0%)	11 (52.3%)	
	missing	4 (2.7%)				
**Nodal status**						0.710
	Negative	44 (29.7%)	13 (26.5%)	25 (33.3%)	6 (28.6%)	
	positive	101 (68.2%)	36 (73.5%)	50 (66.7%)	15 (71.4%)	
	missing	3 (2.0%)				
**Histological grade**						0.049
	G1	11 (7.4%)	6 (12.2%)	5 (6.7%)	0	
	G2	72 (48.6%)	19 (38.8%)	45 (60.0%)	8 (40.0%)	
	G3	61 (41.2%)	24 (49.0%)	25 (33.3%)	12 (60.0%)	
	missing	4 (2.7%)				
**Hormone receptor** **status**						0.252
	negative	23 (15.5%)	11 (22.4%)	9 (11.5%)	3 (14.3%)	
	positive	125 (84.5%)	38 (77.6%)	69 (88.5%)	18 (85.7%)	
**HER2 status**						
	Negative	127 (85.8%)	49 (100.0%)	78 (100.0%)		
	Positive	21 (14.2%)			21 (14.2%)	
	0	49 (33.1%)	49 (100.0%)			
	1+	59 (39.9%)		59 (75.6%)		
	2+	22 (14.9%)		19 (24.4%)	3 (14.3%)	
	2+/ISH negative	19 (12.8%)				
	2+/ISH positive	3 (2.0%)				
	3+	18 (12.2%)			18 (85.7%)	
**Ki-67**						0.022
	<20%	19 (12.8%)	10 (35.7%)	9 (17.3%)	0	
	>20%	74 (50.0%)	18 (64.3%)	43 (82.7%)	13 (100.0%)	
	Missing	55 (37.2%)				
**Molecular subtype**						<0.001
	Luminal-like	107 (72.3%)	38 (77.6%)	69 (88.5%)	0	
	Luminal-A-like	18 (12.2%)	9	9	0	
	Lumina-B-like	46 (31.1%)	10	36	0	
	Missing Ki-67	43 (29.1%)				
	HER2 positive	21 (14.2%)			21 (100.0%)	
	Triple-negative	20 (13.5%)	11 (22.4%)	9 (11.5%)	0	
**Metastatic site**						0.349
	Liver	50 (33.8%)	16 (32.7%)	28 (35.9%)	6 (28.6%)	
	Bone	38 (25.7%)	13 (26.5%)	23 (29.5%)	2 (9.5%)	
	Skin/Soft tissue	18 (12.2%)	6 (12.2%)	10 (12.8%)	2 (9.5%)	
	Central nervous system	15 (10.1%)	6 (12.2%)	4 (5.1%)	5 (23.8%)	
	others	13 (8.8%)	5 (10.2%)	6 (7.7%)	2 (9.5%)	
	Lung/Pleura	9 (6.1%)	1 (2.0%)	5 (6.4%)	3 (14.3%)	
	Lymph node	5 (3.4%)	2 (4.1%)	2 (2.6%)	1 (4.8%)	
**Additional metastatic biopsy**						
	Yes	19 (12.8%)	5 (10.2%)	11 (14.1%)	3 (14.3%)	0.797
	*HER2 concordance with previous biopsy*	8 (42.1%)	1 (20.0%)	6 (54.5%)	1 (33.3%)	
	*HER2 discordance with previous biopsy*	11 (57.9%)	4 (80.0%)	5 (45.4%)	2 (66.7%)	
	No	129 (87.2%)	44 (89.8%)	67 (85.9%)	18 (85.7%)	
**Treatment for early breast cancer**						
	Neo-/Adjuvant chemotherapy	72 (48.6%)	35 (71.4%)	30 (38.5%)	7 (33.3%)	<0.001
	Neo-/Adjuvant Anti-HER2-therapy	9 (6.1%)	0	0	9 (42.9%)	<0.001
	Adjuvant endocrine therapy	79 (53.4%)	31 (63.3%)	42 (56.0%)	6 (28.6%)	0.083
**Treatment for metastatic breast cancer**						
	chemotherapy	45 (30.4%)	8 (16.3%)	26 (34.2%)	11 (52.4%)	0.007
	Anti-HER2-therapy	13 (8.8%)	2 (4.1)	0	11 (52.4%)	<0.001
	Endocrine therapy	52 (35.1%)	18 (36.7%)	29 (38.2%)	5 (23.8%)	0.468
**Tumor progression**						
	Time to metastasis, Median [month]	25 (0–150)	44 (0–150)	14 (0–121)	0 (0–111)	
	Time to metastasis biopsy, Median [month]	39 (0–165)	51 (0–150)	36 (0–165)	17 (0–37)	
	Time from metastasis diagnosis to metastasis biopsy, Median [month]	1 (0–131)	0 (0–55)	1 (0–131)	0 (0–94)	
**Primary metastatic breast cancer (PMBC)**	Yes	53 (35.8%)	8 (16.3%)	31 (39.7%)	14 (66.7%)	
	No	95 (64.9%)	41 (83.7%)	47 (60.3%)	7 (33.3%)	

## Data Availability

The dataset analyzed during the current study is available from the corresponding author upon reasonable request.

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
