# Peer review of "Discordance of HER2-Low between Primary Tumors and Matched Distant Metastases in Breast Cancer"

_cancers, 2023, doi:10.3390/cancers15051413_

Round 1
Reviewer 1 Report
This study examined the discordance of low HER2 levels between primary breast tumor and distant metastases. The authors investigated the differences in HER2 expression between primary tumors and distant metastases, particularly in the group of HER2-low and HER2-zero breast cancer patients. The data presented a relevant discordance of HER2-low status between primary tumors and their corresponding distant metastases. The results were very informative and provide insights into the importance of evaluating potential discordance rates between primary tumor and corresponding distant metastases to HER2-targeted therapy. Two minor comments are listed below.
1. Since primary tumor(s) and distant metastasis were considered simultaneously, it was confused when the authors stated that they examined differences in HER2 expression between primary tumors and distant metastases, particularly within the HER2-negative cohort (HER2-low and HER2-zero). They should have defined the “HER2-negative” term was referring to the primary tumors.
2. Few typos should be corrected, e.g., “2linic-pathological” in line 81. P values (in Table 1 and lines 148, 149, 152, 247, 249, 250, 253, etc.) should be p values.
Author Response
Point-by-point response to the reviewers’ comments
We thank the reviewers for their valuable comments and have prepared a revised version of the manuscript. Enclosed are our responses to the reviewers' comments and suggestions:
Reviewer #1
This study examined the discordance of low HER2 levels between primary breast tumor and distant metastases. The authors investigated the differences in HER2 expression between primary tumors and distant metastases, particularly in the group of HER2-low and HER2-zero breast cancer patients. The data presented a relevant discordance of HER2-low status between primary tumors and their corresponding distant metastases. The results were very informative and provide insights into the importance of evaluating potential discordance rates between primary tumor and corresponding distant metastases to HER2-targeted therapy. Two minor comments are listed below.
Comment 1. Since primary tumor(s) and distant metastasis were considered simultaneously, it was confused when the authors stated that they examined differences in HER2 expression between primary tumors and distant metastases, particularly within the HER2-negative cohort (HER2-low and HER2-zero). They should have defined the “HER2-negative” term was referring to the primary tumors.
Response 1. We absolutely agree with the reviewer that a more precise description of the study population is appropriate. We have clarified the relevant sections in the manuscript, e.g. “We examined differences in HER2 expression between primary tumors and distant metastases, particularly within the HER2-negative primary breast cancer cohort (HER2-low and HER2-zero).”
Comment 2. Few typos should be corrected, e.g., “2linic-pathological” in line 81. P values (in Table 1 and lines 148, 149, 152, 247, 249, 250, 253, etc.) should be p values.
Response 2. We apologize for this inaccuracy. We have changed the noted typos in our manuscript.
Reviewer 2 Report
The work of Katrin Almstedt and coworkers aims to identify differences in HER2 expression between primary tumors and metastases in a large series of breast cancers. In my opinion, the manuscript is largely well structured and documented, and the results are very interesting indeed. I want to congratulate the authors on the presented results and the high value of their manuscript.
However, I would appreciate some explanations and would like the authors to consider my suggestions for improving the manuscript:
The introduction deserves more attention and needs to be amplified. I suggest to the authors a recent publication to overcome this problem (DOI: 10.1159/000524227).
Line 37: it is not clear whether the percentage is also correct with respect to that declared in figure 1. In this regard, I recommend that you review the manuscript in its entirety to be sure.
Table1 “intrinsic subtypes ”: Please use “Her2 enriched”; it is also unclear since the vast majority of luminal carcinomas lack KI67 quantification. This shortcoming must be filled.
Line 104: Please cite Grassini et al. Indeed, there are several unusual patterns of HER2 expression (DOI: 10.1159/000524227).
Author Response
Point-by-point response to the reviewers’ comments
We thank the reviewers for their valuable comments and have prepared a revised version of the manuscript. Enclosed are our responses to the reviewers' comments and suggestions:
Reviewer #2
The work of Katrin Almstedt and coworkers aims to identify differences in HER2 expression between primary tumors and metastases in a large series of breast cancers. In my opinion, the manuscript is largely well structured and documented, and the results are very interesting indeed. I want to congratulate the authors on the presented results and the high value of their manuscript.
However, I would appreciate some explanations and would like the authors to consider my suggestions for improving the manuscript:
Comment 1. The introduction deserves more attention and needs to be amplified. I suggest to the authors a recent publication to overcome this problem (DOI: 10.1159/000524227).
Response 1. We thank the reviewer for the critical evaluation of our introduction and the advice to elaborate it further in light of the review by Grassini et al. We have revised and significantly expanded the introduction accordingly:
“1. Introduction
Overexpression or amplification of human epidermal growth factor receptor 2 (HER2) characterizes a molecular subtype of breast cancer that progresses rapidly and has a poor prognosis [1,2]. However, with the advent of targeted therapies against HER2 such as the monoclonal antibody trastuzumab, the original prognostic disadvantage of HER2 positivity has been transformed into a clinically relevant predictive advantage [3]. In advanced HER2-positive breast carcinoma, survival was further prolonged by pertuzumab [4]. In the event of progression, tyrosine kinase inhibitors such as lapatinib or tucatinib showed efficacy in HER2-positive breast carcinoma [5,6]. In addition, antibody-drug conjugates (ADC) provided a further improvement not only in progression-free survival (PFS) but also in overall survival (OS) in advanced HER2-positive breast carcinoma [7,8].
Therefore, it was obvious to use anti-HER2 therapies also in early breast carcinoma. Indeed, the use of trastuzumab resulted in a statistically significant prolongation of overall survival [9]. Surprisingly, some of the patients who participated in the original trastuzumab trials and were ultimately HER2-negative at central HER2 reassessment benefited from trastuzumab [10,11]. Based on these findings, a large phase III trial was conducted in 3270 women, but it clearly showed that trastuzumab was not beneficial for patients without IHC 3+ or ISH-enhanced breast cancer [12]. In HER2-positive early breast cancer, the addition of pertuzumab led to a relevant increase in pathologic complete response rates (pCR) and an improvement in disease-free survival (DFS) [13,14]. The prolongation of DFS was further increased using the tyrosine kinase inhibitor neratinib after completion of trastuzumab-based therapy in HER2-positive patients [15]. Furthermore, the ADC T-DM1 imroved DFS in early HER2-positve breast cancer with residual disease after neoadjuvant trastuzumab-based treatment [16]. Overall, these HER2-targeted therapies represent tremendous progress for the 15-20% HER2-positive patients. Meanwhile, several retrospective studies have taken a closer look at the large group of HER2-negative breast cancer. Breast cancer showing HER2 protein expression without HER2 gene amplification could be divided into two separate groups (HER2-low [IHC 1+ or 2+ and ISH-negative] and HER2-zero [IHC 0]) with different prognosis or pCR after neoadjuvant chemotherapy [17–22]. However, these results could not be confirmed in several other studies [23–36]. Nevertheless, interest in HER2-low tumors has greatly increased due to the results of the DESTINY-Breast04 trial, which demonstrated the superiority of trastuzumab-deruxtecan (T-DXd) over physician's choice chemotherapy in patients with advanced HER2-low breast cancer [37]. The prolongation of PFS (10.1 months vs 5.4 months; hazard ratio [HR] 0.51; p<0.001) and OS (23.9 months vs 17.5 months; HR 0.64; p=0.003) was both statistically significant and clinically relevant. These compelling results led to a rapid update of the American Society of Clinical Oncology (ASCO) guideline and a positive opinion of the Committee for Medicinal Products for Human Use (CHMP) of the European Medicines Agency (EMA) recommending the use of T-DXd in patients with HER2-low metastatic BC [38,39].
Until now, the only question regarding HER2 status was whether the tumor was HER2-positive or HER2-negative. However, the impressive data from T-DXd in HER2-low breast cancer highlights the importance of dividing the large group of HER2-negative patients. In principle, the well-known ASCO/College of American Pathologists (CAP) clinical practice guideline allows such a distinction [40]. However, potential difficulties such as tumor heterogeneity (clustered or mosaic type) or unusual staining patterns (moderate to intense but incomplete staining or carcinomas with limited strong HER2 overexpression) must be considered. [41]. To address difficulties in distinguishing between HER2-low and HER2-zero, pathologists have already pointed out possible solutions for the assessment of immunohistochemical staining such as (1) application of the "magnification rule", (2) staining pattern-circularity of membrane staining, (3) percentage of tumor cells with HER2 expression [42].
In addition to these briefly outlined challenges for pathologists in distinguishing between HER2-low and HER2-zero, another fundamental problem is discordance between primary tumor and corresponding metastases, since whenever possible, a recent metastatic biopsy is encouraged to guide therapy in advanced breast cancer. The problem of discordance between primary tumor and distant metastases arises when treating patients with metastatic disease with targeted therapies. Discordance of traditional HER2 dichotomy (positive or negative) between primary breast cancer and distant metastases is well established. Among others, Grassini et al. reviewed the phenomenon of HER2 conversion between primary breast tumor and relapsed/distant metastatic [41]. While early studies described a wide variability in HER2 discordance rates (0%-44%), several meta-analyses showed discordance rates ranging from 7.8% to 13.7% . Most commonly, conversion from HER2-positive to HER2-negative was observed, which is clinically important in both advanced and early breast cancer. In neoadjuvant studies, a loss of HER2 expression from therapy-naïve primary tumor and postneoadjuvant residual tumor was described with a prognostic disadvantage [43–48].
However, less is known about the discordance of HER2-low between primary tumor and distant metastases. Thus, Tarantino and coworkers demonstrated a relevant discordance in HER2 expression between PTs and their associated metastases: 44% of HER2-zero PTs had an elevated HER2 score on biopsy, and 22% of HER2-low PTs became HER2-zero tumors [32]. Miglietta et al. reported an overall rate of HER2 discordance of 38.0%, with most transitioning from HER2-zero to HER2-low (15%) and from HER2-low to HER2-zero (14%) [49]. This discordance rate is clinically relevant to the use of ADCs and prompted us to investigate the discordance rate in 148 paired samples (primary breast tumor and distant metastasis), focusing on (i) molecular subtype, (ii) distant metastasis site, and (iii) differences between primary metastatic breast cancer (PMBC) and secondary metastatic breast cancer (SMBC).”
Comment 2. Line 37: it is not clear whether the percentage is also correct with respect to that declared in figure 1. In this regard, I recommend that you review the manuscript in its entirety to be sure.
Response 2. We thank the reviewer for this important comment. We absolutely agree that this statement is confusing because different reference cohorts were chosen. We have now chosen the HER2-negative cohort (n=127) as the reference throughout the manuscript at this point. We have adjusted Figure 1 and the corresponding manuscript sections in Table 1 and in the Discussion.
Figure 1. Patient enrolment.
Comment 3. Table1 “intrinsic subtypes ”: Please use “Her2 enriched”; it is also unclear since the vast majority of luminal carcinomas lack KI67 quantification. This shortcoming must be filled.
Response 3. We thank the reviewer for the careful editing of our manuscript and apologize for the use of the incorrect term "intrinsic subtypes" in Table 1, although we did not use gene expression analyses to determine HER2 status. We have changed the term "intrinsic" to "molecular subtypes."
Indeed, the high number of missing ki-67 values is very regrettable. The reason for this is the recruitment period of our study. The database was searched for patients with metastatic breast cancer between 1995 and 2019, thus including tumors for which ki-67 was not yet part of clinical routine.
Comment 4. Line 104: Please cite Grassini et al. Indeed, there are several unusual patterns of HER2 expression (DOI: 10.1159/000524227).
Response 4. The reviewer is right. We have briefly taken up the problem of unusual patterns in the introduction and have also cited the work here :
“However, potential difficulties such as tumor heterogeneity (clustered or mosaic type) or unusual staining patterns (moderate to intense but incomplete staining or carcinomas with limited strong HER2 overexpression) must be considered [41].”
“HER2-low tumors included all 1+ and 2+ tumors without amplification of HER2. Tumors with a HER2 score 0 were classified as HER2-zero [41,42].”
References
[1] Slamon DJ, Clark GM, Wong SG, Levin WJ, Ullrich A, McGuire WL. Human breast cancer: correlation of relapse and survival with amplification of the HER-2/neu oncogene. Science 1987;235(4785):177–82. https://doi.org/10.1126/science.3798106.
[2] Slamon DJ, Godolphin W, Jones LA, Holt JA, Wong SG, Keith DE et al. Studies of the HER-2/neu proto-oncogene in human breast and ovarian cancer. Science 1989;244(4905):707–12. https://doi.org/10.1126/science.2470152.
[3] Slamon DJ, Leyland-Jones B, Shak S, Fuchs H, Paton V, Bajamonde A et al. Use of chemotherapy plus a monoclonal antibody against HER2 for metastatic breast cancer that overexpresses HER2. N Engl J Med 2001;344(11):783–92. https://doi.org/10.1056/NEJM200103153441101.
[4] Baselga J, Cortés J, Kim S-B, Im S-A, Hegg R, Im Y-H et al. Pertuzumab plus trastuzumab plus docetaxel for metastatic breast cancer. N Engl J Med 2012;366(2):109–19. https://doi.org/10.1056/NEJMoa1113216.
[5] Geyer CE, Forster J, Lindquist D, Chan S, Romieu CG, Pienkowski T et al. Lapatinib plus capecitabine for HER2-positive advanced breast cancer. N Engl J Med 2006;355(26):2733–43. https://doi.org/10.1056/NEJMoa064320.
[6] Murthy RK, Loi S, Okines A, Paplomata E, Hamilton E, Hurvitz SA et al. Tucatinib, Trastuzumab, and Capecitabine for HER2-Positive Metastatic Breast Cancer. N Engl J Med 2020;382(7):597–609. https://doi.org/10.1056/NEJMoa1914609.
[7] Cortés J, Kim S-B, Chung W-P, Im S-A, Park YH, Hegg R et al. Trastuzumab Deruxtecan versus Trastuzumab Emtansine for Breast Cancer. N Engl J Med 2022;386(12):1143–54. https://doi.org/10.1056/NEJMoa2115022.
[8] Verma S, Miles D, Gianni L, Krop IE, Welslau M, Baselga J et al. Trastuzumab emtansine for HER2-positive advanced breast cancer. N Engl J Med 2012;367(19):1783–91. https://doi.org/10.1056/NEJMoa1209124.
[9] Trastuzumab for early-stage, HER2-positive breast cancer: a meta-analysis of 13 864 women in seven randomised trials. The Lancet Oncology 2021;22(8):1139–50. https://doi.org/10.1016/S1470-2045(21)00288-6.
[10] Paik S, Kim C, Wolmark N. HER2 status and benefit from adjuvant trastuzumab in breast cancer. N Engl J Med 2008;358(13):1409–11. https://doi.org/10.1056/NEJMc0801440.
[11] Perez EA, Reinholz MM, Hillman DW, Tenner KS, Schroeder MJ, Davidson NE et al. HER2 and chromosome 17 effect on patient outcome in the N9831 adjuvant trastuzumab trial. J Clin Oncol 2010;28(28):4307–15. https://doi.org/10.1200/JCO.2009.26.2154.
[12] Fehrenbacher L, Cecchini RS, Geyer CE JR, Rastogi P, Costantino JP, Atkins JN et al. NSABP B-47/NRG Oncology Phase III Randomized Trial Comparing Adjuvant Chemotherapy With or Without Trastuzumab in High-Risk Invasive Breast Cancer Negative for HER2 by FISH and With IHC 1+ or 2. J Clin Oncol 2020;38(5):444–53. https://doi.org/10.1200/JCO.19.01455.
[13] Gianni L, Pienkowski T, Im Y-H, Tseng L-M, Liu M-C, Lluch A et al. 5-year analysis of neoadjuvant pertuzumab and trastuzumab in patients with locally advanced, inflammatory, or early-stage HER2-positive breast cancer (NeoSphere): a multicentre, open-label, phase 2 randomised trial. Lancet Oncol 2016;17(6):791–800. https://doi.org/10.1016/S1470-2045(16)00163-7.
[14] Minckwitz G von, Procter M, Azambuja E de, Zardavas D, Benyunes M, Viale G et al. Adjuvant Pertuzumab and Trastuzumab in Early HER2-Positive Breast Cancer. N Engl J Med 2017;377(2):122–31. https://doi.org/10.1056/NEJMoa1703643.
[15] Chan A, Delaloge S, Holmes FA, Moy B, Iwata H, Harvey VJ et al. Neratinib after trastuzumab-based adjuvant therapy in patients with HER2-positive breast cancer (ExteNET): a multicentre, randomised, double-blind, placebo-controlled, phase 3 trial. The Lancet Oncology 2016;17(3):367–77. https://doi.org/10.1016/S1470-2045(15)00551-3.
[16] Minckwitz G von, Huang C-S, Mano MS, Loibl S, Mamounas EP, Untch M et al. Trastuzumab Emtansine for Residual Invasive HER2-Positive Breast Cancer. N Engl J Med 2019;380(7):617–28. https://doi.org/10.1056/NEJMoa1814017.
[17] Li Y, Abudureheiyimu N, Mo H, Guan X, Lin S, Wang Z et al. In Real Life, Low-Level HER2 Expression May Be Associated With Better Outcome in HER2-Negative Breast Cancer: A Study of the National Cancer Center, China. Front Oncol 2021;11:774577. https://doi.org/10.3389/fonc.2021.774577.
[18] Mutai R, Barkan T, Moore A, Sarfaty M, Shochat T, Yerushalmi R et al. Prognostic impact of HER2-low expression in hormone receptor positive early breast cancer. Breast 2021;60:62–9. https://doi.org/10.1016/j.breast.2021.08.016.
[19] Denkert C, Seither F, Schneeweiss A, Link T, Blohmer J-U, Just M et al. Clinical and molecular characteristics of HER2-low-positive breast cancer: pooled analysis of individual patient data from four prospective, neoadjuvant clinical trials. Lancet Oncol 2021;22(8):1151–61. https://doi.org/10.1016/S1470-2045(21)00301-6.
[20] Almstedt K, Heimes A-S, Kappenberg F, Battista MJ, Lehr H-A, Krajnak S et al. Long-term prognostic significance of HER2-low and HER2-zero in node-negative breast cancer. Eur J Cancer 2022;173:10–9. https://doi.org/10.1016/j.ejca.2022.06.012.
[21] Rosso C, Voutsadakis IA. Characteristics, Clinical Differences and Outcomes of Breast Cancer Patients with Negative or Low HER2 Expression. Clin Breast Cancer 2022;22(4):391–7. https://doi.org/10.1016/j.clbc.2022.02.008.
[22] Tan, Ryan Shea Ying Cong, Ong WS, Lee K-H, Lim AH, Park S, Park YH et al. HER2 expression, copy number variation and survival outcomes in HER2-low non-metastatic breast cancer: an international multicentre cohort study and TCGA-METABRIC analysis. BMC Med 2022;20(1):105. https://doi.org/10.1186/s12916-022-02284-6.
[23] Tarantino P, Niman SM, Erick TK, Priedigkeit N, Harrison BT, Giordano A et al. HER2-low inflammatory breast cancer: Clinicopathologic features and prognostic implications. Eur J Cancer 2022;174:277–86. https://doi.org/10.1016/j.ejca.2022.07.001.
[24] Chen M, Chen W, Liu D, Chen W, Shen K, Wu J et al. Prognostic values of clinical and molecular features in HER2 low-breast cancer with hormonal receptor overexpression: features of HER2-low breast cancer. Breast Cancer 2022. https://doi.org/10.1007/s12282-022-01364-y.
[25] Douganiotis G, Kontovinis L, Markopoulou E, Ainali A, Zarampoukas T, Natsiopoulos I et al. Prognostic Significance of Low HER2 Expression in Patients With Early Hormone Receptor Positive Breast Cancer. Cancer Diagn Progn 2022;2(3):316–23. https://doi.org/10.21873/cdp.10111.
[26] Horisawa N, Adachi Y, Takatsuka D, Nozawa K, Endo Y, Ozaki Y et al. The frequency of low HER2 expression in breast cancer and a comparison of prognosis between patients with HER2-low and HER2-negative breast cancer by HR status. Breast Cancer 2021. https://doi.org/10.1007/s12282-021-01303-3.
[27] Jacot W, Maran-Gonzalez A, Massol O, Sorbs C, Mollevi C, Guiu S et al. Prognostic Value of HER2-Low Expression in Non-Metastatic Triple-Negative Breast Cancer and Correlation with Other Biomarkers. Cancers (Basel) 2021;13(23). https://doi.org/10.3390/cancers13236059.
[28] Agostinetto E, Rediti M, Fimereli D, Debien V, Piccart M, Aftimos P et al. HER2-Low Breast Cancer: Molecular Characteristics and Prognosis. Cancers (Basel) 2021;13(11). https://doi.org/10.3390/cancers13112824.
[29] Schettini F, Chic N, Brasó-Maristany F, Paré L, Pascual T, Conte B et al. Clinical, pathological, and PAM50 gene expression features of HER2-low breast cancer. NPJ Breast Cancer 2021;7(1):1. https://doi.org/10.1038/s41523-020-00208-2.
[30] Shao Y, Yu Y, Luo Z, Guan H, Zhu F, He Y et al. Clinical, Pathological Complete Response, and Prognosis Characteristics of HER2-Low Breast Cancer in the Neoadjuvant Chemotherapy Setting: A Retrospective Analysis. Ann Surg Oncol 2022. https://doi.org/10.1245/s10434-022-12369-4.
[31] Shu L, Tong Y, Li Z, Chen X, Shen K. Can HER2 1+ Breast Cancer Be Considered as HER2-Low Tumor? A Comparison of Clinicopathological Features, Quantitative HER2 mRNA Levels, and Prognosis among HER2-Negative Breast Cancer. Cancers (Basel) 2022;14(17). https://doi.org/10.3390/cancers14174250.
[32] Tarantino P, Gandini S, Nicolò E, Trillo P, Giugliano F, Zagami P et al. Evolution of low HER2 expression between early and advanced-stage breast cancer. Eur J Cancer 2022;163:35–43. https://doi.org/10.1016/j.ejca.2021.12.022.
[33] van den Ende, Nadine S, Smid M, Timmermans A, van Brakel JB, Hansum T, Foekens R et al. HER2-low breast cancer shows a lower immune response compared to HER2-negative cases. Sci Rep 2022;12(1):12974. https://doi.org/10.1038/s41598-022-16898-6.
[34] Won HS, Ahn J, Kim Y, Kim JS, Song J-Y, Kim H-K et al. Clinical significance of HER2-low expression in early breast cancer: a nationwide study from the Korean Breast Cancer Society. Breast Cancer Res 2022;24(1):22. https://doi.org/10.1186/s13058-022-01519-x.
[35] Xu H, Han Y, Wu Y, Wang Y, Li Q, Zhang P et al. Clinicopathological Characteristics and Prognosis of HER2-Low Early-Stage Breast Cancer: A Single-Institution Experience. Front Oncol 2022;12:906011. https://doi.org/10.3389/fonc.2022.906011.
[36] Gampenrieder SP, Dezentjé V, Lambertini M, Nonneville A de, Marhold M, Le Du F et al. Influence of HER2 expression on prognosis in metastatic triple-negative breast cancer-results from an international, multicenter analysis coordinated by the AGMT Study Group. ESMO Open 2022;8(1):100747. https://doi.org/10.1016/j.esmoop.2022.100747.
[37] Modi S, Jacot W, Yamashita T, Sohn J, Vidal M, Tokunaga E et al. Trastuzumab Deruxtecan in Previously Treated HER2-Low Advanced Breast Cancer. N Engl J Med 2022;387(1):9–20. https://doi.org/10.1056/NEJMoa2203690.
[38] Moy B, Rumble RB, Carey LA. Chemotherapy and Targeted Therapy for Human Epidermal Growth Factor Receptor 2-Negative Metastatic Breast Cancer That Is Either Endocrine-Pretreated or Hormone Receptor-Negative: ASCO Guideline Rapid Recommendation Update. J Clin Oncol 2022:JCO2201533. https://doi.org/10.1200/JCO.22.01533.
[39] ESMO. EMA Recommends Extension of Therapeutic Indications for Trastuzumab Deruxtecan. European Society for Medical Oncology (ESMO) 2022, 22 December 2022; Available from: https://www.esmo.org/oncology-news/ema-recommends-extension-of-therapeutic-indications-for-trastuzumab-deruxtecan2. [January 04, 2023].
[40] Wolff AC, Hammond MEH, Allison KH, Harvey BE, Mangu PB, Bartlett JMS et al. Human Epidermal Growth Factor Receptor 2 Testing in Breast Cancer: American Society of Clinical Oncology/College of American Pathologists Clinical Practice Guideline Focused Update. J Clin Oncol 2018;36(20):2105–22. https://doi.org/10.1200/JCO.2018.77.8738.
[41] Grassini D, Cascardi E, Sarotto I, Annaratone L, Sapino A, Berrino E et al. Unusual Patterns of HER2 Expression in Breast Cancer: Insights and Perspectives. Pathobiology 2022;89(5):278–96. https://doi.org/10.1159/000524227.
[42] Denkert C, Lebeau A, Schildhaus HU, Jackisch C, Rüschoff J. New treatment options for metastatic HER2-low breast cancer Consequences for histopathological diagnosis. Pathologie (Heidelb) 2022. https://doi.org/10.1007/s00292-022-01139-4.
[43] Guarneri V, Dieci MV, Barbieri E, Piacentini F, Omarini C, Ficarra G et al. Loss of HER2 positivity and prognosis after neoadjuvant therapy in HER2-positive breast cancer patients. Ann Oncol 2013;24(12):2990–4. https://doi.org/10.1093/annonc/mdt364.
[44] Mittendorf EA, Wu Y, Scaltriti M, Meric-Bernstam F, Hunt KK, Dawood S et al. Loss of HER2 amplification following trastuzumab-based neoadjuvant systemic therapy and survival outcomes. Clin Cancer Res 2009;15(23):7381–8. https://doi.org/10.1158/1078-0432.CCR-09-1735.
[45] Tural D, Karaca M, Zirtiloglu A, M Hacioglu B, Sendur MA, Ozet A. Receptor discordances after neoadjuvant chemotherapy and their effects on survival. J BUON 2019;24(1):20–5.
[46] Wang R-X, Chen S, Jin X, Chen C-M, Shao Z-M. Weekly paclitaxel plus carboplatin with or without trastuzumab as neoadjuvant chemotherapy for HER2-positive breast cancer: loss of HER2 amplification and its impact on response and prognosis. Breast Cancer Res Treat 2017;161(2):259–67. https://doi.org/10.1007/s10549-016-4064-9.
[47] Ignatov T, Gorbunow F, Eggemann H, Ortmann O, Ignatov A. Loss of HER2 after HER2-targeted treatment. Breast Cancer Res Treat 2019;175(2):401–8. https://doi.org/10.1007/s10549-019-05173-4.
[48] Branco FP, Machado D, Silva FF, André S, Catarino A, Madureira R et al. Loss of HER2 and disease prognosis after neoadjuvant treatment of HER2+ breast cancer. Am J Transl Res 2019;11(9):6110–6.
[49] Miglietta F, Griguolo G, Bottosso M, Giarratano T, Lo Mele M, Fassan M et al. Evolution of HER2-low expression from primary to recurrent breast cancer. NPJ Breast Cancer 2021;7(1):137. https://doi.org/10.1038/s41523-021-00343-4.
Round 2
Reviewer 2 Report
The recommended changes have been largely edited.
Publication in this form is recommended